# Integrative Transcriptomics and Proteomics Analysis Reveals Immune Response Process in Bovine Viral Diarrhea Virus-1-Infected Peripheral Blood Mononuclear Cells

**DOI:** 10.3390/vetsci10100596

**Published:** 2023-09-28

**Authors:** Kang Zhang, Jingyan Zhang, Lei Wang, Qiang Liang, Yuhui Niu, Linlin Gu, Yanming Wei, Jianxi Li

**Affiliations:** 1College of Veterinary Medicine, Gansu Agricultural University, Lanzhou 730070, China; 15719319192@163.com (K.Z.); zhaoban@gsau.edu.cn (L.W.);; 2Lanzhou Institute of Husbandry and Pharmaceutical Sciences, Chinese Academy of Agricultural Sciences, Lanzhou 730050, China; 3College of Veterinary Medicine, Shandong Vocational Animal Science and Veterinary College, Weifang 261061, China; 4Shenzhen Bioeasy Biotechnology Co., Ltd., Shenzhen 518100, China; gull@bioeasy.com

**Keywords:** BVDV, transcriptomics, proteomics, peripheral blood mononuclear cells

## Abstract

**Simple Summary:**

Bovine viral diarrhea virus (BVDV) is a viral pathogen that poses a significant threat to the global cattle industry. BVDV infection interferes with the host’s innate and adaptive immunity, thereby affecting the diverse organs and systems of the body. Thus, comprehending the biological mechanisms by which viruses infect hosts is pivotal for disease control and prevention. Currently, the molecular mechanisms of the cell signaling pathways affected by BVDV infection remain unclear. Transcriptomic and proteomic methods offer promising avenues to dissect the biological processes in virus-infected host cells. In this study, transcriptomics and proteomics techniques were used to analyze differentially expressed proteins and scrutinize differentially expressed genes in immune-related biological processes after BVDV-1 infection in cow peripheral blood mononuclear cells (PBMCs). Our findings indicate that BVDV-1 predominantly suppresses the host through pathways such as the interleukin-17 signaling pathway, cytokine-cytokine receptor interaction, and complement and coagulation cascades. Conversely, PBMCs seem to counter viral invasion through mechanisms such as complement and coagulation cascades, the RIG-I-like receptor signaling pathway, and cytokine-cytokine receptor interaction. This study forms a theoretical basis for analyzing the mechanism of BVDV-1 infection and facilitates a deeper exploration of host responses to BVDV-1 infection in dairy cows.

**Abstract:**

Bovine viral diarrhea virus (BVDV) causes bovine viral diarrhea-mucosal disease, inflicting substantial economic losses upon the global cattle industry. Peripheral blood mononuclear cells (PBMCs) are the central hub for immune responses during host-virus infection and have been recognized as crucial targets for BVDV infection. In order to elucidate the dynamics of host-BVDV-1 interaction, this study harnessed RNA-seq and iTRAQ methods to acquire an extensive dataset of transcriptomics and proteomics data from samples of BVDV-1-infected PBMCs at the 12-h post-infection mark. When compared to mock-infected PBMCs, we identified 344 differentially expressed genes (DEGs: a total of 234 genes with downregulated expression and 110 genes with upregulated expression) and 446 differentially expressed proteins (DEPs: a total of 224 proteins with downregulated expression and 222 proteins with upregulated expression). Selected DEGs and DEPs were validated through quantitative reverse transcriptase-polymerase chain reaction and parallel reaction monitoring. Gene ontology annotation and KEGG enrichment analysis underscored the significant enrichment of DEGs and DEPs in various immunity-related signaling pathways, including antigen processing and presentation, complement and coagulation cascades, cytokine-cytokine receptor interaction, and the NOD-like receptor signaling pathway, among others. Further analysis unveiled that those DEGs and DEPs with downregulated expression were predominantly associated with pathways such as complement and coagulation cascades, the interleukin-17 signaling pathway, cytokine-cytokine receptor interaction, the PI3K-Akt signaling pathway, the tumor necrosis factor signaling pathway, and the NOD-like receptor signaling pathway. Conversely, upregulated DEGs and DEPs were chiefly linked to metabolic pathways, oxidative phosphorylation, complement and coagulation cascades, and the RIG-I-like receptor signaling pathway. These altered genes and proteins shed light on the intense host-virus conflict within the immune realm. Our transcriptomics and proteomics data constitute a significant foundation for delving further into the interaction mechanism between BVDV and its host.

## 1. Introduction

Bovine viral diarrhea virus (BVDV) type A, a positive-sense single-stranded RNA virus, belongs to the genus *Pestivirus* in the family Flaviviridae [1]. BVDVs can be classified into three major groups based on genetic diversity and antigenic differential analysis: BVDV-1, BVDV-2, and BVDV-3 (HoBi-like virus) [2,3]. BVDV infection is marked by symptoms such as diarrhea, respiratory diseases, and immunosuppression, leading to reduced performance in the cattle industry and substantial economic losses [4,5]. Animals with persistent infection (PI) carry the virus for an extended period and intermittently release the virus into the environment, which serves as the primary source of BVDV transmission [6]. BVDV-1 and BVDV-2 have begun to spread among various livestock species in China, such as cattle [7], pigs [8], and goats [9]. Our previous studies also further revealed an increase in BVDV infection in large western Chinese dairy farms, with the predominant subtypes being BVDV-1a, BVDV-1m, and BVDV-1q [10]. Thus, comprehensive exploration of PI cattle detection and eradication holds immense significance for the prevention and control of bovine viral diarrhea-mucosal disease in cattle herds.

BVDV replication is a complex and intricate process that occurs within cellular cytoplasm. BVDV particles initially bind to designated cell receptors. Subsequently, this is followed by the fusion of the viral envelope with the cell membrane in a pH-dependent manner [11]. The host cell surface may bear numerous BVDV receptors, with CD46 being one of the potential cell receptors for BVDV [12]. The virus-host interaction triggers robust resistance from the host immune system. Innate immunity, the initial defense against foreign pathogens, plays a crucial role in restraining viral replication. BVDV utilizes various tactics to elude the natural immune system, which encompasses impeding the synthesis of interferon (IFN), dampening the protein activity of IFN, and disrupting the expression or operation of IFN-induced antiviral effector proteins [13]. BVDV can infect immune cells, prompting apoptosis. For instance, when BVDV infects peripheral blood lymphocytes under in vitro conditions, the depletion of B cells, helper T cells, and cytotoxic T cells to varying degrees has been observed [14,15]. BVDV also significantly affects monocytes, with highly virulent BVDV causing a reduction in calf peripheral blood mononuclear cells (PBMCs) by 30–70% [16]. BVDV also leads to diminished phagocytosis in alveolar macrophages, the downregulated expression of immunoglobulin Fc receptor and complement receptor (C3R), and decreased chemokine levels [17]. The nuclear factor-κB (NF-κB) pathway, a major route for pro-inflammatory signaling, undergoes strong and rapid activation and inhibition upon BVDV infection. While BVDV infection promptly stimulates the NF-κB signaling pathway (15 min post-infection), it also triggers the overexpression of the negative regulators of this signaling cascade, effectively impeding its activation [18,19]. A recent study indicated that BVDV-1 infection increases the factor A20 level, resulting in NF-κB activity blockage and approximately an 80% reduction in IL-8 expression [20]. The host’s interaction mechanism with BVDV and the pathogenesis of BVDV infection exhibit an intricate landscape, where the foundation of BVDV PI lies in the evasion of the host’s immune defense system. Despite extensive research endeavors, the exact mechanisms governing viral replication, pathogenesis, and host innate immunity evasion remain mysterious, particularly in the initial stages of viral infection. Recent years have witnessed the identification of potential biomarkers that are pivotal for virus-host interactions by analyzing the differential transcriptomic data of hosts before and after virus infection [21]. Using an RNA-seq transcriptomic method, the analysis of Vero cells infected with PEDV highlighted noteworthy alterations in the mTOR signaling pathway at various time intervals following viral infection. Subsequent experiments further validated its significance in virus proliferation [22]. A cellular-level approach holds the potential to unlock the intricacies of how BVDV persists in infecting host cells. Shifts in host transcriptome expression profiles during BVDV infection point to the repression of antiviral genes in the complement system as potential factors influencing BVDV proliferation [23]. A recent study demonstrated that the BVDV-2 infection of goat PBMCs prompts a series of transcriptional changes in the DEGs associated with the immune responses of the host and covers inflammation, defense responses, and cytokine/chemokine-mediated signaling transduction, among others [24]. However, it is important to note that transcriptomics data solely reflect mRNA expression levels, not the existence of post-translation modification, which is not enough for prediction. As such, predicting protein expression levels solely from quantitative mRNA data alone is insufficient. In order to explore the early-stage biological processes of interaction between BVDV and host immune systems, this study embarked on obtaining transcriptome and protein profiles of BVDV-1-infected dairy cow PBMCs, integrating transcriptomics and proteomics, followed by a comprehensive analysis. This research establishes a theoretical foundation for unraveling the mysteries of BVDV infection within host cells.

## 2. Materials and Methods

### 2.1. Ethics Approval

All experimental procedures involving animals were approved by the Animal Ethics Committee of the Lanzhou Institute of Husbandry and Pharmaceutical Sciences of CAAS (permission number: SYXK (Gan) 2019-0002).

### 2.2. Virus and Cells

The Madin-Darby bovine kidney cells were obtained from the China Institute of Veterinary Drug Control and were cultured in Dulbecco’s modified Eagle’s medium (Hyclone, Logan, UT, USA). The medium was supplemented with 10% fetal bovine serum (FBS; TransGen, Biotech, Co., Ltd., Beijing, China) and maintained at 37 °C. To generate stock virus, an MDBK cell monolayer was inoculated with cpBVDV-1a (GenBank No. OR187364, isolated and maintained in our laboratory [10]) with a multiplicity of infection of 0.1. After 3 days of infection, the cells that were affected underwent harvesting. These cells were then lysed through three consecutive freeze-thaw cycles. Following that, they were centrifuged at a speed of 8000× *g* and a temperature of 4 °C for a duration of 10 min. The resulting samples were further partitioned into smaller portions and preserved at a temperature of −70 °C until they were ready for utilization.

### 2.3. Dairy Cow, PBMC Culture, and Viral Infection

Blood sample collection was performed on 3-year-old Holstein cows from Lanzhou Manor Ranch in Gansu Province. Before selection, these cows were carefully screened to confirm their freedom from BVDV-specific antibodies and antigens. EDTA anticoagulated whole-blood samples were gathered from four cows. The collected blood (*n* = four cows) was mixed in a 1:1 ratio with phosphate-buffered saline (PBS) and diluted in a large cell culture bottle. Subsequently, the isolation and culture of PBMCs were performed using the following methodology: First, 4 mL of lymphatic separation solution was added to the bottom of a sterile 15 mL centrifuge tube. Subsequently, 8 mL of the diluted blood was gently added along the tube wall, ensuring clear stratification. This solution mixture was centrifuged at 3000 rpm for 20 min, yielding distinct stratification. The stratified layers observed included a plasma layer (containing PBS) at the top, a lymphoid layer with white rings, a separation layer, and an erythrocyte layer. The white ring-shaped lymphocyte layer was then carefully collected into a new centrifuge tube. After adding an appropriate amount of cell washing solution, this mixture solution was centrifuged at 2000 rpm for 10 min, followed by supernatant removal. Furthermore, 5 mL of the red blood cell lysate was added and centrifuged at 2000 rpm for 10 min, followed by supernatant removal. Subsequent washing steps were performed at 2000 rpm for 10 min [25]. The resulting multi-tubule cells were combined with preheated RPMI 1640 medium containing 10% FBS [26]. Cell viability was counted using Sigma-Aldrich staining and was observed under a light microscope. The incubator temperature was set at 37 °C with a CO_2_ concentration of 5%. Cells were then cultured in six-well plates containing 3 mL of medium in each well, adjusting the cell density to 3 × 10^6^ cells/mL. A blank control group was also included (cells cultured with 1640 medium containing 10% FBS) and a BVDV-positive group. The final virus concentration was set at 10^2^ by adding the virus stock solution to the cell culture according to TCID_50_. The virus group cells were cultured separately from the non–virus group cells. Cells were cultured for various time points (0, 3, 6, 9, 12, and 24 h post-infection [hpi.]), and culture supernatants were collected and stored at −80 °C for subsequent analysis.

### 2.4. RNA Extraction, Sequencing, and Bioinformatics Analyses

Total RNA was extracted and purified from BVDV-1-infected PBMCs and control PBMCs using TRIzol reagent (15596018; Thermo Fisher Scientific, New York, NY, USA), according to the manufacturer’s instructions (four biological replicates per group). The Bioanalyzer 2100 and RNA 6000 Nano LabChip Kit (Agilent, Santa Clara, CA, USA) were used to analyze total RNA quantity and purity. The RNA samples with an RIN number of >7.0 were chosen for constructing the sequencing library. After the extraction of total RNA, mRNA was purified in two rounds from total RNA using Dynabeads Oligo (dT) (Thermo Fisher Scientific, Santa Clara, CA, USA). After purification, the mRNA underwent fragmentation into brief fragments through the use of divalent cations at a high temperature (cat. e6150; Magnesium RNA Fragmentation Module, New England Biolabs, Ipswich, Ma, USA) at 94 °C for 5–7 min. Then, the fragmented RNA pieces were subjected to reverse transcription utilizing SuperScript™ II Reverse Transcriptase (cat. 1896649; Invitrogen, Carlsbad, CA, USA) in order to produce cDNA. The obtained cDNA was utilized to synthesize second-stranded DNAs labeled with U by employing *E. coli* DNA polymerase I (cat. m0209; New England Biolabs, Ipswich, Ma, USA), RNase H (cat. m0297; New England Biolabs, Ipswich, MA, USA), and a solution containing dUTP (cat. R0133; Thermo Fisher Scientific, USA). Subsequently, An A-base was introduced to the blunt ends of each strand to facilitate their ligation to the indexed adapters. Each adapter was designed to possess a T-base overhang to ensure successful ligation to the A-tailed fragmented DNA. The fragments were then ligated with dual-index adapters, which were subsequently subjected to size selection utilizing AMPureXP beads. Following treatment with a heat-labile UDG enzyme (cat. m0280; New England Biolabs, Ipswich, Ma, USA), the ligated products underwent amplification through PCR using the following conditions: an initial denaturation step at 95 °C for 3 min, succeeded by eight cycles consisting of denaturation at 98 °C for 15 s, annealing at 60 °C for 15 s, extension at 72 °C for 30 s, and a final extension at 72 °C for 5 min. The resulting cDNA library exhibited an average insert size of 300 ± 50 bp. Lastly, Illumina NovaSeq™ 6000 was employed to conduct 2 × 150-bp paired-end sequencing (PE150), following the recommended protocol provided by the vendor.

After sequencing, we removed unqualified reads, which included sequencing adapters and low-quality sequences, and obtained rRNA-mapped reads. Consequently, these resulting reads were aligned to the reference genome using Hisat2 (version: hisat2-2.2.1, https://daehwankimlab.github.io/hisat2/, accessed on 7 December 2022). We normalized gene expression levels using the fragments per kilobase million approach based on the normalization of the original read count of the gene. Genes exhibiting |log_2_FC| > 1 and q < 0.05 were classified as significantly differentially expressed genes (DEGs). Afterward, the DEGs underwent an analysis in the gene ontology (GO) database to provide functional annotation, as well as in the Kyoto Encyclopedia of Genes and Genomes (KEGG) database to identify pathways enriched significantly. GO enrichment analysis detected GO terms that showed considerable enrichment in DEGs compared with the genome background, allowing for the filtering of DEGs related to specific biological functions. KEGG, a valuable resource for comprehensive gene function analysis, was used to identify metabolic pathways or signal transduction pathways that displayed significant enrichment and involved the DEGs when compared to the whole-genome background. Moreover, all of the original sequence data have been submitted to the NCBI gene expression omnibus datasets and assigned the accession number (accession No. PRJNA 1003857).

### 2.5. Validation of RNA-Seq Results by RT-qPCR

To ensure the credibility of the RNA-seq sequencing data, a quantitative reverse transcription-PCR (RT-qPCR) analysis was performed to authenticate the DEGs detected from transcriptome sequencing. For validation purposes, ten DEGs were randomly chosen from the RNA-seq sequencing data: NOS2, TNFAIP3, CTSL, MMP9, CSF3, IL1R2, CCL20, TNFRSF13B, F3, and CXCR2. These DEGs played key roles in various signaling pathways, including antigen processing and presentation, interaction between cytokines and cytokine receptors, the calcium signaling pathway, PI3K-Akt signaling pathway, complement and coagulation cascades, NF-κB signaling pathway, the TNF signaling pathway, the Toll-like receptor signaling pathway, and the IL-17 signaling pathway.—PBMC samples infected with BVDV were prepared according to the previously mentioned method, while control samples consisted of PBMC samples without any infection. The total RNA was extracted and subjected to SYBR Green-based qRT-PCR using Roche SYBR qPCR Master Mix (Basel, Switzerland) and the primers listed in Table 1. The qRT-PCR analysis was carried out using the ABI 7500 system (Applied Biosystems, Foster City, CA, USA), with the ACTB gene serving as an internal reference gene. The 2^−ΔΔCt^ method was applied to determine the relative expression levels of the target genes. Each qRT-PCR assay was performed in four technical replicates.

### 2.6. Protein Extraction and iTRAQ Labelling

PBMCs infected with BVDV and mock-infected PBMCs were prepared in accordance with the previously described procedure. Cells were collected from each group using a cell scraper, and there were four biological replicates for each group. Subsequently, the cells were sonicated in a lysis buffer consisting of 1% sodium dodecyl sulfate, 1% sodium deoxycholate, 1% NP40, 15 mM NaCl, 25 mM Tris-HCl, and a complete protease inhibitor cocktail (Roche, Nutley, NJ, USA) at 4 °C for 20 min. Following the sonication, the cell lysates were centrifuged at 12,000 rpm for 10 min at 4 °C, and the resulting supernatant was collected. The total protein concentration was determined using a Bicinchoninic Acid Protein Assay Kit (Abcam, Fremont, CA, USA). 

Afterward, each sample underwent overnight digestion with modified sequence-grade trypsin (Promega, San Luis Obispo, CA, USA) at 37 °C, using 100 μg of protein. Subsequently, the dissolved digested contents were incorporated into 500 mM tetraethylammonium bromide (Sigma, Washington, DC, USA). Eventually, the peptides obtained were labeled with the iTRAQ 8-plex reagent kit (SCIEX, Redwood City, CA, USA) and left to react for 2 h before undergoing vacuum drying.

### 2.7. High pH Prefractionation and Nano-LC-MS/MS Analysis

After vacuum drying, the polypeptides were re-dissolved in buffer A (10 mM aqueous solution of ammonium acetate, pH 10) at 50 °C. Subsequently, separation was carried out using RPUPLC (150 mm × 2.1 mm; Waters, XBridge BEH C18 XP) under alkaline conditions. A linear gradient of buffer B (10 mM ammonium acetate, 10% H_2_O, 90% acetonitrile [ACN], pH 10) ranging from 2% to 90% was utilized over 60 min, followed by high-pH separation (maintaining a flow rate of 0.3 mL/min and a temperature of 30 °C). Individual components were collected and subsequently vacuum-dried. For each sample, a 2-μL total peptides sample was subjected to separation and analyzed by a nano-UPLC (EASY-nLC1200) coupled to a Q Exactive HFX Orbitrap instrument (Thermo Fisher Scientific) equipped with a nano-electrospray ion source. To conduct the experiment, we utilized a reversed-phase column (Reprosil-Pur 120 C18-AQ, 1.9 μm, Dr. Maisch) measuring 100 μm ID × 15 cm. The column allowed for efficient separation of our samples. The mobile phases employed in the separation process consisted of two solutions: H_2_O containing 0.1% FA and 2% ACN (phase A) and phase B consisting of 80% ACN and 0.1% FA. To achieve separation, a gradient lasting 90 min was applied at a flow rate of 300 nL/min. The gradient was designed as follows: B: 2–5% (2 min), 5–22% (68 min), 22–45% (16 min), 45–95% (2 min), and 95% (2 min). The execution of data-dependent acquisition (DDA) followed a profile and positive mode approach, employing an Orbitrap analyzer. The resolution for MS1 was set at 120,000 (@200 *m*/*z*), and the *m*/*z* range was specified as 350–1600. In the case of MS2, a resolution of 45,000 was selected while maintaining a fixed first mass of 110 *m*/*z*. To ensure optimal control, the automatic gain control (AGC) target for MS1 was established at 3×10^6^ with a maximum integration time (IT) of 30 ms. Similarly, for MS2, the AGC target was set to 1E5 with a maximum IT of 96 ms. Fragmentation of the top 20 most intense ions occurred through higher-energy collisional dissociation (HCD) technique with a normalized collision energy (NCE) of 32%. The isolation window for HCD was determined to be 0.7 *m*/*z*. Additionally, a dynamic exclusion time window of 45 s was implemented, encompassing both single-charged peaks and peaks exceeding a charge of six, thereby excluding them from the DDA process.

### 2.8. Protein Bioinformatics Analysis

The mass spectrometry data were employed by Proteome Discoverer (PD) software (Version 2.4.0.305) and its integrated Sequest HT search engine. The UniProt FASTA databases at the species level (uniprot-Bos + taurus (Bovine)-9913-2020-10) were employed to conduct searches of the MS spectra lists. Both the PSM and peptide levels had the false discovery rate (FDR) set at 0.01. Peptide identification involved a deviation of up to 10 ppm for the initial precursor mass and 0.02 Da for the fragment mass. Protein quantification utilized a unique peptide and Razor peptide, while the total peptide amount was used for normalization. Quantitative analysis was conducted on proteins with at least 2 unique spectra, with a significance threshold of *p*-value < 0.05 to identify differentially expressed proteins (DEP). Subsequently, the DEPs underwent GO database analysis and KEGG database analysis to annotate their functional properties and enrich their associated pathways, respectively. All original sequence reads were submitted to the NCBI database as a BioProject (PRJNA1003857).

### 2.9. Validation of iTRAQ Results by Parallel Reaction Monitoring

Fourteen DEPs were randomly selected for targeted quantification by parallel reaction monitoring (PRM) conducted by Biotree Biotech Co., Ltd., (Shanghai, China). The peptides utilized for PRM analysis were synthesized following the established protocol used for TMT. To conduct a DDA mode test, the PRM samples were loaded onto a trap column (100 μm × 2 cm, C18, 3 μm, 150 Å) at a flow rate of 5 μL/min for a duration of 10 min. Next, each individual sample underwent separation on a C18 reversed-phase column (PePSep C18, 1.9 μm, 75 μm × 25 cm; Bruker, Karlsruhe, Germany) using a linear gradient. For optimal mass resolution, the primary setting was adjusted to 60,000, with the AGC target value being set at 4e5. The maximum ion injection time was defined as 50 ms. Full MS scans were acquired within the mass range of 100–1700 *m*/*z*. Subsequently, MS/MS fragmentation was achieved through higher-energy collisional dissociation (HCD) utilizing an NCE value of 35. Additional settings for MS/MS included a resolution of 15,000 and an AGC target value of 5×10^4^. The maximum ion injection time was once again set to 50 ms, while dynamic exclusion was established at 15.0 s to prevent repeated fragment analysis. In order to perform PRM mode testing, the parameter “chromatography in the DDA PaSEF mode” was selected. To integrate the desired peptide lists obtained from DDA results into the inclusion list of the Xcalibur PRM method editing module, Skyline software was employed. The following PRM parameters were configured: an MS2 resolution of 15,000, an isolation window of 1.6 *m*/*z*, an AGC value of 5×10^4^, a maximum accumulated time of 60 ms, and an HCD energy of 35. Both the control and test groups were thoroughly evaluated using four independent biological replicates. 

### 2.10. Statistical Analysis 

All data were presented as mean with standard error of the mean. Comparisons between the two groups were performed using Student’s t-test. Differences were considered statistically significant when *p* < 0.05 or q < 0.05. Statistical analysis was performed using SPSS v.22.0 software (SPSS Inc., Chicago, IL, USA).

## 3. Results

### 3.1. Determination of BVDV-1 Replication in Cow PBMCs

In order to assess the growth characteristics of BVDV on PBMCs, we utilized RT-PCR and qRT-PCR techniques to validate the replication of BVDV-1 in cow PBMCs. As depicted in Figure 1A, fragments of the 5′-UTR (~300 bp) were amplified in the cow PBMCs infected with BVDV-1 over a period of 0 to 24 hpi. The results of qRT-PCR showed a continuous increase in the BVDV genome copy number with infection time, peaking at 12 hpi. (Figure 1B). These findings indicated BVDV-1 infection in cow PBMCs, with no BVDV nucleotides detected in PBMCs mock-infected after 24 hpi. In order to delve into the mechanisms of BVDV-1-PBMC interactions, we collected BVDV-1-infected PBMC samples at 12 hpi for RNA-seq-based transcriptomics and iTRAQ-based proteomics analyses.

### 3.2. Quality Evaluation of the Transcriptome and Differentially Expressed Gene Analysis

Transcriptome sequencing of control and BVDV-1-infected PBMCs was conducted using an Illumina NovaSeq^™^ 6000 platform. After RNA sequencing, we obtained 323,896,256 raw reads (NC: 170,172,594; V: 153,723,662). After removing the low-quality reads and reads with adapter sequences, 314,906,984 valid reads were obtained (NC: 165,372,268; V: 149,534,716; Table 2). These valid reads were aligned to the most recent reference genome using “http://tophat.cbcb.umd.edu/ (accessed on 7 December 2022)”. Regarding the NC and V samples, the reference genome obtained 156,345,812 and 141,183,185 mapped reads, respectively. The corresponding mapping rates for both samples were 94.54% and 94.41% (Table 2). When comparing the mock-infected PBMCs group with the BVDV-1-infected PBMC group, we identified 344 DEGs by using a false discovery rate of q < 0.05 and |log_2_FC| > 1 (Figure 2A). Among these, 110 had significantly upregulated expression, and 234 had significantly downregulated expression. A volcano plot visualizing the DEG distribution was also generated (Figure 2B). Appendix A presents the details of all DEGs. Reference genome “ftp://ftp.ensembl.org/pub/release-107/fasta/bos_taurus/dna/” (accessed on 7 December 2022).

For a deeper understanding of the biological functions of the DEGs, GO annotation analyses were conducted. In the biological process category, most DEGs were associated with processes such as the G protein-coupled receptor signaling pathway, signal transduction, proteolysis, protein phosphorylation, and the positive regulation of cell migration (Figure 3A). Notably, some DEGs were involved in the negative regulation of apoptosis, positive regulation of ERK1 and ERK2 cascades, and immune responses. In the cellular component category, the largest number of DEGs was primarily involved in the integral component of the membrane, plasma membrane, and cytoplasm. Particularly, DEGs with upregulated expression were enriched in the cytosol, nucleus, and mitochondrion, possibly indicating the use of cellular components by BVDV-1 for viral replication. Molecular function analysis revealed that the DEGs were associated with binding and G-protein-coupled receptor activity. Additional enrichment analysis using GO demonstrated that the DEGs were significantly enriched in various pathways, such as extracellular matrix, transmembrane receptor protein tyrosine kinase signaling, virus receptor activity, plasma membrane, interleukin-8 binding, the positive regulation of interleukin-10 production, and the positive regulation of nitric oxide synthase activity (Figure 3B). The DEGs with downregulated expression were enriched in immune-related GO terms, suggesting that BVDV-1 may hinder host innate immunity by suppressing the host immune system and apoptosis during the early stages of infection.

In order to gain further insights into DEG functions, a KEGG pathway enrichment analysis was conducted. Among the top 20 enriched pathways (Figure 4A), extracellular matrix (ECM)-receptor interaction, cytokine-cytokine receptor interaction, complement and coagulation cascades, cell adhesion molecules, antigen processing and presentation, and the calcium signaling pathway were closely related to immune responses. An analysis of the significant differences revealed that pathways such as antigen processing and presentation, ECM-receptor interaction, complement and coagulation cascades, cell adhesion molecules, the calcium signaling pathway, cytokine-cytokine receptor interaction, the PI3K-Akt signaling pathway, the TNF signaling pathway, and the IL-17 signaling pathway were significantly enriched with DEGs (Figure 4B and Appendix A). Notably, the cytokine-cytokine receptor interaction pathway showed a high number of enriched DEGs (n = 11). Among these DEGs, TNFRSF13B, IFNLR1, IL12B, and TNFRSF12A had upregulated expression, whereas IL1R2, CSF3, CXCR2, CXCR1, CCL20, TGFB2, and NGFR had downregulated expression (Appendix A). Further analysis indicated that the DEGs with downregulated expression were primarily enriched in pathways such as cell adhesion molecules, complement and coagulation cascades, the calcium signaling pathway, the IL-17 signaling pathway, cytokine-cytokine receptor interaction, the PI3K-Akt signaling pathway, the Rap1 signaling pathway, the MAPK signaling pathway, and the TNF signaling pathway (Appendix A). Conversely, DEGs with upregulated expression were mainly enriched in pathways such as the cAMP signaling pathway, antigen processing and presentation, sulfur metabolism, the RIG-I-like receptor signaling pathway, cytokine-cytokine receptor interaction, ABC transporters, and ECM-receptor interaction. Most of these pathways play pivotal roles in regulating host immunity. Antigen processing and presentation, in particular, is strongly associated with immune evasion. Our findings provide critical foundational information to gain a better understanding of BVDV-1-host interactions at the mRNA level (Appendix A).

### 3.3. Quality Evaluation of the Proteome and Analysis of Different Proteins

Mock-infected PBMCs were used as the negative control, and proteome sequencing was conducted on BVDV-1-infected PBMCs by using iTRAQ, followed by LC-MS/MS analysis. The DEPs were identified by comparing them with the mock-infected PBMC group based on the criteria of FC ≤ 0.83 or FC ≥ 1.2 and *p* < 0.05. The results presented in Figure 5A reveal 446 DEPs, including 222 with upregulated expression and 224 with downregulated expression. Additionally, a volcano plot illustrating the distribution of DEPs was generated (Figure 5B).

Following this, GO annotation enrichment analysis was conducted for these DEPs. The biological process categories of DEPs, as depicted in Figure 6A, suggested that most DEPs were primarily related to the cellular component organization or biogenesis, cellular component organization, protein-containing complex assembly, and more. Several DEPs were notably involved in the stress response, positive regulation of stimulus-response, immune response regulation, and integrin-mediated signaling pathway (Appendix A). Notably, in regard to most GO terms, particularly those associated with the host’s innate immunity, the number of DEPs exhibiting downregulated expression surpassed that of DEPs displaying upregulated expression. This implies that BVDV has the potential to impede the host’s immune system during the early stages of viral infection, consequently improving its own survival. Conversely, when considering the term related to metabolic processes, the number of DEPs with upregulated expression significantly outweighed the count of DEPs with downregulated expression. This signifies that BVDV can exploit the host’s metabolism to facilitate viral replication. This discovery aligns with the outcomes of the transcriptome analysis. 

In order to delve into the biological functions of the DEPs, a KEGG pathway enrichment analysis was conducted. The outcomes highlighted that all DEPs significantly enriched 34 KEGG pathways. Notably, Complement and coagulation cascades and Oxidative phosphorylation emerged as the two most enriched and significant pathways (Figure 6B). Intriguingly, certain DEPs showed significant enrichment in the pathways related to immune regulation, such as Complement and coagulation cascades, NOD-like receptor signaling pathway, ECM-receptor interaction, and PPAR signaling pathway. A more comprehensive KEGG pathway enrichment analysis was carried out on all DEPs (Appendix A). The results demonstrated that the DEPs with downregulated expression were primarily enriched in pathways such as focal adhesion, complement and coagulation cascades, ECM-receptor interaction, Fc gamma R-mediated phagocytosis, the cGMP-PKG signaling pathway, and the NOD-like receptor signaling pathway. Meanwhile, those DEPs with upregulated expression were mainly enriched in metabolic pathways, oxidative phosphorylation, complement and coagulation cascades, and peroxisome. These findings provide essential context for understanding BVDV-1-PBMC interactions at the protein level.

### 3.4. Verification of DEGs by qRT-PCR and DEPs by Parallel Reaction Monitoring

In order to further corroborate the RNA-Seq data, qRT-PCR analysis was performed on selected DEGs associated with immune responses. The results (Table 3) demonstrated that 10 chosen genes exhibited consistent directionality between RNA-Seq and qRT-PCR analysis, with a high correlation coefficient (R^2^ = 0.94). This validates the reliability of the DEGs identified through RNA-Seq. Similarly, the PRM analysis of DEPs confirmed the abundance levels of 14 DEPs, which aligned with the iTRAQ proteomics analysis (Appendix A). 

### 3.5. Combined Analysis of Transcriptome and Proteome Data

In order to achieve a comprehensive understanding of BVDV-1 infection of PBMCs, a joint analysis of transcriptome and proteome data was conducted. The correlation analysis focused on the DEGs and DEPs significantly enriched in immune-related GO terms and KEGG pathways. As illustrated in Figure 7, HPX and NLRP3, which are enriched in the regulation of immune response GO terms, exhibited a positive correlation with the DEGs primarily involved in cytokine-cytokine receptor interaction and the IL-17 signaling pathway (e.g., CSF3, TNFAIP3, CCL20, CXCR2). Conversely, they showed a negative correlation with CTSL and TNFRSF13B. TFRC, CD40, CD74, and TBK1, which were closely related to the NOD-like receptor signaling pathway and NF-κB signaling pathway, were positively correlated with DEGs enriched in antigen processing and presentation (e.g., CTSL, TNFRSF13B, and NOS2) and negatively correlated with DEGs such as F3/CSF3 and TNFAIP3. Moreover, C9, C4BPA, CARD9, IRF1, and APOA1 were predominantly enriched in pathways such as complement and coagulation cascades, the NOD-like receptor signaling pathway, and the C-type lectin receptor signaling pathway, which exhibited similar relationships with these DEGs.

## 4. Discussion

BVDV causes acute bovine viral diarrhea as well as PI, leading to substantial economic losses in the global cattle industry [4]. A meta-analysis revealed a BVDV-positive rate of 27.1% in Chinese dairy herds, underlining the significant threat posed by BVDV to the dairy sector [7]. BVDV can induce immunosuppression by impacting the host immune system, thereby evading immune defenses, promoting viral survival, and exacerbating subsequent infections [27]. Studies on infection mechanisms have highlighted that BVDV initially targets innate immune cells such as epithelial cells, dendritic cells, monocytes, and macrophages, contributing to immune system disruption in infected animals [28]. Despite some scholarly investigations into the interaction between BVDV and the host, detailed insights into viral replication, pathogenesis, and natural host immune evasion mechanisms remain elusive. Transcriptomics and proteomics techniques allow us to explore biological processes at the mRNA and protein levels, offering the potential for a systematic analysis of virus–host interactions. While studies have examined mRNA expression changes in different cell types upon the BVDV infection of cattle and goats [19,23,24], some scholars have delved into transcriptomic and proteomic data from BVDV-infected MDPK cells [29]. However, the majority of studies have focused on a single type of omics data, analyzing the changes in DEGs and DEPs following the BVDV infection of host cells. In order to comprehensively study the early-stage interaction between BVDV and the host, we infected bovine PBMCs with BVDV-1, collected cell samples at 12 hpi, analyzed the transcriptome and proteome, and systematically investigated the changes in differential genes and proteins to shed light on the early response mechanism of BVDV-1 virus infection in PBMCs.

Innate immunity serves as the frontline defense against antigen invasion and plays a vital role in the host’s resistance to viral infections. Viral nucleic acids and proteins act as essential pathogen-associated molecular patterns that can be recognized by pattern recognition receptors (PRRs). This recognition process triggers pathways such as the Toll-like signaling pathway and the RIG-I signaling pathway, resulting in a surge of antiviral cytokines [30,31]. PBMCs and various cell subsets serve as sentinels against viral intrusion, playing a pivotal role in both innate and adaptive immune responses [32]. Following the viral invasion, cell PRRs, particularly RIG-I-like receptors, detect viral RNA in the cytoplasm, activating immune responses. RIG-I, equipped with a caspase activation and recruitment domain (CARD), is coupled with mitochondrial antiviral signaling proteins, thereby triggering the activation of interferon regulatory factors (IRF-3 and IRF-7) and NF-κB in the nucleus [33]. This cascade culminates in the activation of antiviral genes, including interferons, interferon-stimulated genes, and pro-inflammatory factors, effectively curbing viral replication and spread. Our RNA-seq data analysis indicated a significant increase in the expression of IL12B and ISG15; both are associated with the RIG-I-like receptor signaling pathway. This suggests that BVDV-1 is recognized by RIG-I receptors upon the infection of PBMCs, thus initiating innate immunity. Notably, ISG15 is a crucial antiviral protein that is pivotal in the ubiquitin-like modification system that contributes to the innate immune response, proving essential for the antiviral effects of interferon [34,35]. IFN-λ, which is known to inhibit respiratory and gastrointestinal mucosal viral infections such as HIV [36], has been recently shown to suppress BVDV infection as well. Treatment of ncp-BVDV-2-infected calves with recombinant bovine IFN-λ resulted in undetectable virus levels in nasal secretions and serum and elevated IFN-I levels [37,38]. Our study observed a substantial increase in the expression of IFNLR1, a member of the IFN-λ receptor family. In contrast, a study on BVDV-infected MDBK cells demonstrated the downregulation of host antiviral protein expression in the early stages of infection [29]. The IL-17 family, recognized for its inflammatory cytokines, significantly contributes to innate and adaptive immunity. The IL-17 family comprises many members, and although they all play a role in promoting the inflammatory response, they all have their own unique functions in response to the invasion of different pathogens [39,40]. The present study shows that the CSF3, CCL20, MMP3, and MMP9 genes enriched in the IL-17 signaling pathway are significantly downregulated during BVDV-1 infection, which was confirmed by the results of qRT-PCR analysis. Matrix metalloproteins (MMPs) are a family of zinc-dependent endopeptidases that play a wide range of physiological functions in the body. MMPs promote cell proliferation, migration, differentiation, and tissue repair and play an important role in immune response and tumor invasion and metastasis. The genetic changes associated with innate immunity suggest a complex interaction between BVDV and the early stages of infection in the host, with the host and virus engaging in an intense battle between immune activation and immune suppression.

Upon pathogen invasion, the innate immune response is the first line of action. Inflammatory cytokines play a pivotal role in activating innate immune cells, initiating complement activation, and triggering specific immune responses. Inflammatory cytokines such as IFN, IL, and tumor necrosis factor (TNF) activate specific signal transduction pathways to realize their biological functions. Our study revealed that the DEGs altered during BVDV-1 infection of PBMCs were prominently enriched in the cytokine-cytokine receptor interaction KEGG pathway. Of the 11 DEGs involved in this pathway, seven genes (TGFB2, CCL20, NGFR, CXCR1, CXCR2, IL1R2, and CSF3) showed significantly downregulated expression, whereas four genes showed significantly upregulated expression (TNFRSF13B, IFNLR1, TNFRSF12A, and IL12B). These findings align with studies conducted on BVDV-infected MDBK cells, where DEGs were primarily related to cytokine-cytokine receptor interaction, with the downregulated genes associated with immune processes, cell death, and stimuli [29]. Similar trends were identified in BVDV-2-infected goat PBMCs, with the enriched DEGs linked to cytokine-cytokine receptor interactions, TNF signaling pathways, and chemokine signaling pathways [24]. Chemokines, which are low-molecular-weight (8–10 kDa) pro-inflammatory cytokines, orchestrate leukocyte recruitment, cell proliferation, differentiation, and cell death [41,42]. Chemokines play a pivotal role in co-ordinating inflammatory responses, lymphocyte migration, and immune cell differentiation, playing an important role in the development of innate and adaptive immunity. In response to the viral invasion, STING-TBK1/IKKε activates STAT6, which subsequently regulates the expression of various chemokines, including CCL20 [43]. CCL20 can function as an inflammatory and homeostatic chemokine, and its role in allergic airway disease is well documented. Our results, however, indicated a significant downregulation of CCL20 [44], which was corroborated by the results of the qRT-PCR data. CXCR1 and CXCR2, which are also members of the chemokine family that are predominantly expressed on neutrophils, play a critical role in acute inflammation [45]. IL-1 enhances natural killer cell activity, exhibits chemotactic effects, and augments neutrophil, macrophage, and lymphocyte functions, primarily regulating immune responses and mediating inflammation [46]. IL-1R2, expressed by neutrophils during tissue recruitment following inflammatory stimuli, may contribute to the suppression and elimination of acute inflammation [47]. CSF3, a protective cytokine with anti-inflammatory effects, is a key regulator of neutrophil production and is crucial for clearing bacterial pathogens and modulating inflammatory responses [48]. The strength of the host defense inflammatory response during bacterial infection is directly related to CSF3 receptor (CSF3R, or CD114/G-CSFR) signaling [49]. The significantly downregulated expression of these cytokines and chemokines indicated that BVDV-1 was immunosuppressive to the host. ncp-BVDV infection increases the expression of CXCR4 and CXCL12 mRNA in bovine PBMCs [50]. Another study investigated the chemokine expression profile of cp- and ncp-BVDV-infected bovine monocytes/macrophages. The results showed that several key chemokines of the CCL and CXCL families were upregulated by cp-BVDV but not by ncp-BVDV [51]. A recent study showed that BVDV-2 infection resulted in the increased transcription of CCL3, CCL4, CCL5, CCL20, and CXCL10 and the decreased transcription of CCL2 in goat PBMCs [24]. The different results may be related to the type of BVDV and are also closely related to the time of infection, and the exact reasons need to be further explored. Notably, the expression of the TNFRSF13B, IFNLR1, TNFRSF12A, and IL12B genes was notably upregulated in BVDV-1-infected PBMCs, which was supported by qRT-PCR validation. TNFRSF13B and TNFRSF12A, which are members of the TNF-like receptor family, regulate immune cell survival and apoptosis and control B-cell antibody responses and immunoglobulin production [52]. Sazzini found that TNFRSF13B might also be associated with innate immunity, expanding beyond adaptive immunity roles [53]. IFNLR1, a type II cytokine receptor, interacts with type III IFN (IFNλs) to engage in signal transduction, inducing antiviral and antitumor functions. IFNλs regulate immunity while conferring antiviral properties, contributing to reduced susceptibility to viruses in collaboration with IFNLR1 [54]. The fluctuating expression of these functional genes suggests a fierce immunological clash between BVDV-1 and host cells during the early infection stages, though the precise mechanisms warrant further investigation.

The complement system, comprising over 35 soluble plasma proteins, constitutes a crucial element of innate defense, bridging innate and adaptive immune responses and actively preventing infections [55,56]. Our analysis of the transcriptome and proteome data showed that both DEGs and DEPs were markedly enriched in the complement and coagulation cascades pathway, with qRT-PCR and PRM validation confirming these findings. In the context of BVDV-1 infection of PBMCs, the expression of the F3, F13A1, ENSBTAG00000023026, and C5 genes experienced significant downregulation. The complement system is integral to innate immunity; upon activation, C5 undergoes cleavage into C5a and C5b. C5b, along with C6, C7, C8, and C9, forms the membrane attack complex to remove invading pathogens, which is crucial for pathogen elimination [56]. Notably, our transcriptomic data showed the prevalent downregulation of those genes related to the complement system. Conversely, the proteomic analysis indicated that six genes were enriched in the complement and coagulation cascade pathways with both up- and downregulated expression. This observation suggests some limitations in our study due to its sole reliance on a single methodology. The changes in genes and proteins associated with the complement system imply that BVDV-1 inhibits this system, playing a pivotal role in the initial stages of BVDV infection.

## 5. Conclusions

In summary, the analysis of RNA-Seq and iTRAQ data revealed a series of immune-related DEGs and DEPs in the peripheral blood of BVDV-1-infected dairy cows. The study’s findings underscore that BVDV-1 primarily suppresses and evades the host through pathways like the IL-17 signaling pathway, cytokine-cytokine receptor interaction, the PI3K-Akt signaling pathway, complement and coagulation cascades, the TNF signaling pathway, and the NOD-like receptor signaling pathway, ultimately achieving PI. Concurrently, the host seems to activate innate immunity against viral invasion through pathways such as complement and coagulation cascades, the RIG-I-like receptor signaling pathway, and cytokine-cytokine receptor interaction. These results deepen our understanding of BVDV-host interactions during BVDV-1 infection and offer novel insights for comprehending the intricate dynamics of host-virus survival mechanisms.

## Figures and Tables

**Figure 1 vetsci-10-00596-f001:**
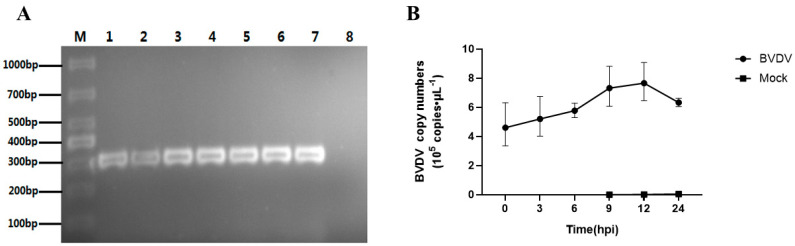
Identification of viral infection in the peripheral blood mononuclear cells (PBMCs) of dairy cows. (**A**) Amplification of 5′ UTR by quantitative reverse transcription-polymerase chain reaction at 0, 3, 6, 9, 12, and 24 hpi in BVDV-1-infected bovine PBMCs. M: DNA marker DL-1000 plus; 1: Infected PBMCs at 0 hpi; 2: Infected PBMCs at 3 hpi; 3: Infected PBMCs at 6 hpi; 4: Infected PBMCs at 9 hpi; 5: Infected PBMCs at 12 hpi; 6: Infected PBMCs at 24 hpi; 7: positive control; 8: mock-infected PBMCs at 24 hpi. (**B**) Detection of viral genome copy numbers in cow PBMCs using qRT-PCR at different time points.

**Figure 2 vetsci-10-00596-f002:**
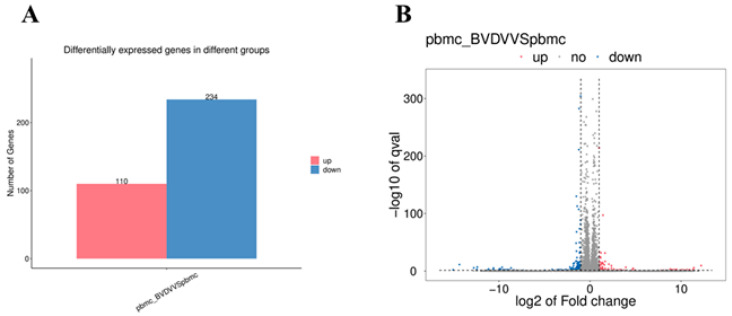
(**A**) Statistical data for the DEGs according to the RNA-seq-based transcriptomics data for BVDV−1-infected PBMCs compared with mock-infected PBMCs. (**B**) Volcano plot of the DEGs.

**Figure 3 vetsci-10-00596-f003:**
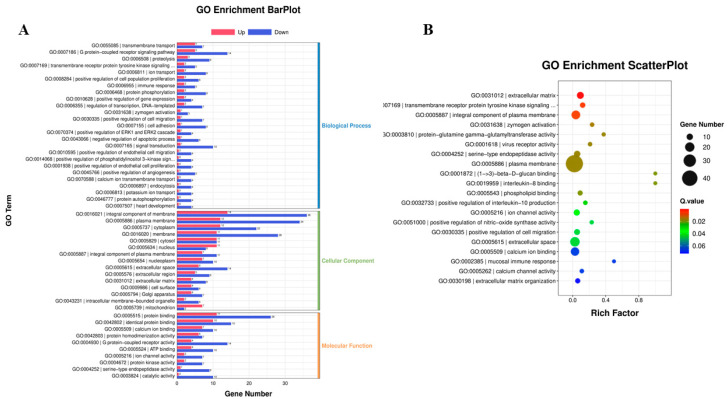
(**A**) Gene ontology (GO) annotation for the differentially expressed genes (DEGs) between mock and BVDV-1-infected cow peripheral blood mononuclear cells (PBMCs). (**B**) Volcano plot of the DEGs. (**B**) GO enrichment analysis for the DEGs between the mock and BVDV-1-infected cow PBMCs. The circles indicate the number of enriched genes, and the colors depict the *q*-value.

**Figure 4 vetsci-10-00596-f004:**
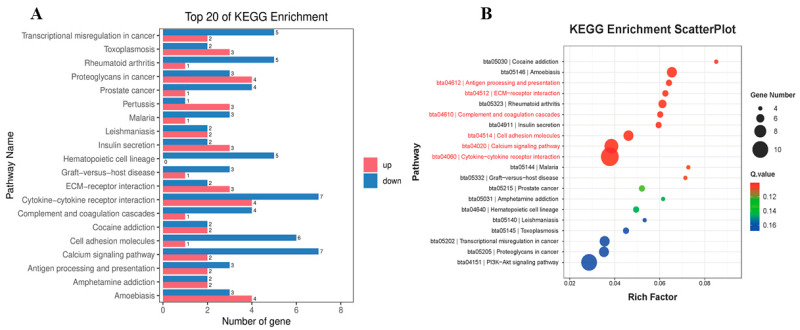
(**A**) Top 20 Kyoto Encyclopedia of Genes and Genomes (KEGG) enrichment for the differentially expressed genes (DEGs) between mock and BVDV-1-infected cow peripheral blood mononuclear cells (PBMCs). (**B**) Volcano plot of the DEGs. (**B**) KEGG pathway enrichment analysis of the DEGs in the RNA-seq transcriptomics data of BVDV-1-infected PBMCs are compared with the mock-infected PBMCs. The circles indicate the number of enriched genes, and the colors depict the *q*-value. The pathways marked in red were immune-related pathways.

**Figure 5 vetsci-10-00596-f005:**
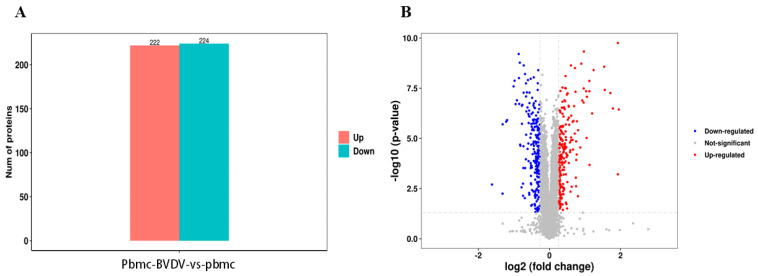
(**A**) Statistics for the differentially expressed proteins (DEPs) according to the TMT-iTRAQ proteomics data for BVDV-1-infected peripheral blood mononuclear cells (PBMCs) when compared with mock-infected PBMCs. (**B**) Volcano plot of the DEPs.

**Figure 6 vetsci-10-00596-f006:**
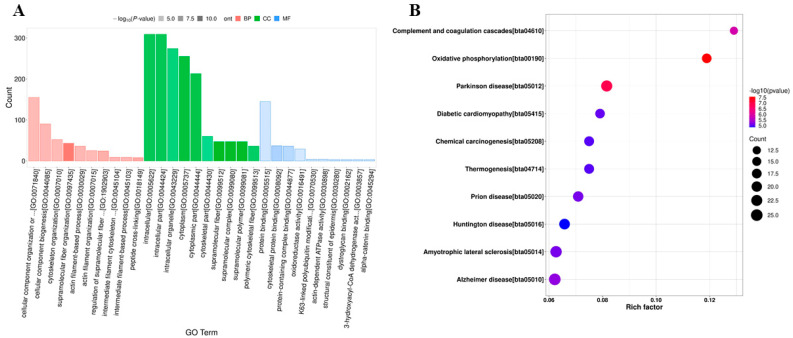
(**A**) Gene ontology annotation for the differentially expressed proteins (DEPs) between mock- and BVDV-1-infected cow peripheral blood mononuclear cells (PBMCs). (**B**) Kyoto Encyclopedia of Genes and Genomes pathway enrichment analysis of the DEPs of BVDV-1-infected PBMCs compared with mock-infected PBMCs. The circles indicate the number of enriched proteins, and the colors depict the *q*-value.

**Figure 7 vetsci-10-00596-f007:**
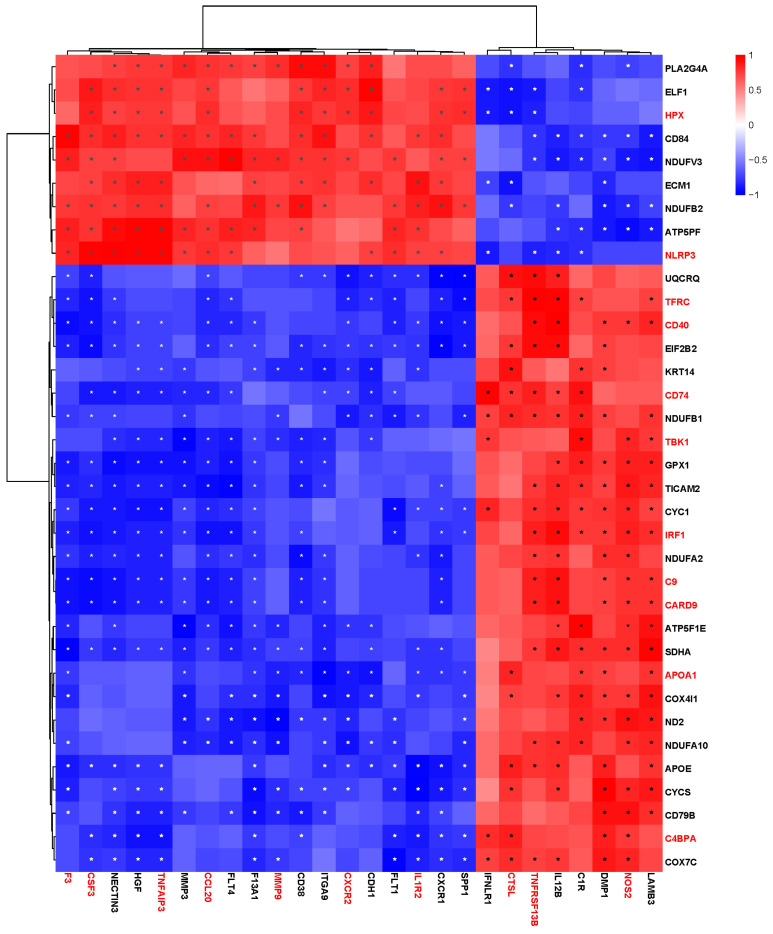
Correlation heatmap of differentially expressed genes (DEGs) and differentially expressed proteins (DEPs) of BVDV-1-infected peripheral blood mononuclear cells (PBMCs), as compared with mock-infected PBMCs. The abscissa represents DEGs, whereas the ordinate represents DEPs. The ones marked in red are DEGs or DEPs that can be significantly enriched in immune-related KEGG pathways. Red indicates a positive correlation, and blue indicates a negative correlation. The darker the color, the stronger the correlation. At the same time, significant correlations were marked with asterisks (*p* < 0.05). The DEPs or DEGs with similar expression levels are clustered.

**Table 1 vetsci-10-00596-t001:** Primers used in qRT-PCR validation.

Genes	Primer Sequences (5′-3′)	PCR Product (bp)
n.ACTB	F: TCTGGCACCACACCTTCTACAAC	168 bp
	R: GATACCCATCTCCGTGCTCTCTAAC	
MMP9-F	F: GACGCCGCTCACCTTCACTC	86 bp
	R: GATACCCATCTCCGTGCTCTCTAAC	
CXCR2-F	F: CATGCTGTTCTGCTACGGATTCAC	142 bp
	R: CACGATCAGGACCAGGTTGTAGG	
PTX3-F	F: GTGGGTGGTGGCTTTGATGAAAC	168 bp
	R: GTGGGGCTGAATCTCTGTGACTC	
CTSL-F	F: CAGGCACACGATGAATGGCTTTC	156 bp
	R: GCCCAACAAGAACCACATTTACCC	
TNFAIP3-F	F: ACAATGAGCAGGGACGGAGAG	112 bp
	R: ATGAAGAATGGGCAGTTAGGTGTC	
CSF3-F	F: ACGAGCTGCCTGAACCAACTAC	140 bp
	R: CAGATGTTCGTGGCAAAGTCAGTG	
IL1R2-F	F: CATGGAGGACGCAGGCTACTATAC	155 bp
	R: GTGAGGCTGAGATGGTCTGGTG	
CCL20-F	F: ACTTCGACTGCTGTCTCCGATATAC	104 bp
	R: AACTGCATTGATGTCACAGGCTTC	
SF13B-F	F: CCAAGAGCAAGGCAGGTATTATGAC	114 bp
	R: CTCCTCAGCGTCTTCTCACAGTAG	
F3-F	F: GGCTCTTCTATTCGGCTTAGTCCTC	142 bp
	R: GACATGATTGATGGGTTTGGGTTCC	
NOS2-F	F: GGTACGAATGGTTCCGGGAG	121 bp
	R: CCCATGTACCACCCGTTGAA	

**Table 2 vetsci-10-00596-t002:** The results of the quality control of the transcriptomics data.

Sample	Raw Data	Valid Data	Mapped Reads	Valid Ratio (Reads)	Q20%	Q30%	GC Content%
NC_R1	43,792,662	42,534,790	40,081,237 (94.23%)	97.13	99.97	97.04	45.50
NC_R2	44,614,414	43,345,270	40,999,375 (94.59%)	97.16	99.97	97.11	45.50
NC_R3	37,510,276	36,461,870	34,464,721 (94.52%)	97.21	99.97	97.25	45.50
NC_R4	44,255,242	43,030,338	40,800,479 (94.82%)	97.23	99.97	97.23	45.50
V_R1	37,922,494	36,873,992	34,804,716 (94.39%)	97.24	99.97	97.13	47.00
V_R2	36,932,852	35,938,950	33,921,839 (94.39%)	97.31	99.97	97.08	47.00
V_R3	37,957,180	36,945,006	34,816,947 (94.24%)	97.33	99.97	97.06	47.00
V_R4	40,911,136	39,776,768	37,639,683 (94.63%)	97.23	99.97	97.13	47.00

NC: mock-infected PBMC group, V: BVDV-1-infected PBMC group.

**Table 3 vetsci-10-00596-t003:** Validation results of RNA-sequencing data using qRT-PCR.

Gene_id	Gene_name	FC log_2_(V/NC)		Regulation
		RNA-Seq	qRT-PCR	
ENSBTAG00000006894	NOS2	1.00	1.50	up
ENSBTAG00000000436	TNFAIP3	−0.43	−0.19	down
ENSBTAG00000000720	CTSL	1.06	0.85	up
ENSBTAG00000020676	MMP9	−1.00	−0.68	down
ENSBTAG00000021462	CSF3	−1.03	−1.06	down
ENSBTAG00000006343	IL1R2	−1.87	−1.48	down
ENSBTAG00000021326	CCL20	−1.43	−1.64	down
ENSBTAG00000015298	TNFRSF13B	1.35	1.15	up
ENSBTAG00000007101	F3	−1.23	−1.04	down
ENSBTAG00000038042	CXCR2	−1.52	−0.99	down

NC: mock-infected PBMC group, V: BVDV-1-infected PBMC group.

## Data Availability

The figures and tables contain the data that support the presented findings and conclusions. Our samples’ raw data have been stored in the NCBI. You can find the raw data under BioProject, Accession Number: PRJNA1003857. The corresponding author can provide additional research materials and protocols upon reasonable request, which are pertinent to the investigation.

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
