# Peer review of "Integrative Transcriptomics and Proteomics Analysis Reveals Immune Response Process in Bovine Viral Diarrhea Virus-1-Infected Peripheral Blood Mononuclear Cells"

_vetsci, 2023, doi:10.3390/vetsci10100596_

Round 1
Reviewer 1 Report
The authors evaluate the dynamics of host-BVDV-1 interaction from a transcriptomic and proteomic approaches. This study shows an innovate direction trough the analysis of virus-host interaction based on the correlation between DEG and PEG analysis. The MS is well-written and organized, but some points can be improved.
Strengths:
The promising results and exhaustive description of the immune genes that appear regulated in the infection condition.
Minor comments:
1. Please italicize “p” in "p < 0.05" in the manuscript.
2. Substitute h.p.i. by hpi.
3. Material and methods: Could you please clarify if you used DNAse treatment in the procedure of 2.4 and 2.5?
4. Material and methods: Can you confirm if you used only one housekeeping gene and why? Next time take into account to use at least two for a more accurate analysis.
5. Results: Please correct the quality of the figures.
6. Results: Why in the figure 1B does the mock condition appears only from 9 hpi onwards? And substitute Time(hpi) by Time (hpi).
7.Results: Why is q < -0.5 used as a criterion for DEGs? And FC<0.83 for DEPs?
8. Table 3: Fix symbols (Log2 and -)
Line 48: Substitute Interleu-kin-17 by Interleukin-17
Line 49: Space after interaction is missing.
Line64-66: Rewrite the sentence because it is not clear.
Line 163: Space after 12 is missing.
Author Response
Response to reviewer 1
Dear reviewer 1,
Thank you very much for taking the time to put forward some helpful suggestions for my manuscript. I have carefully revised the manuscript in accordance with your suggestions, the details are as follows.
Best regards,
Sincerely,
Kang Zhang
General comments:
The authors evaluate the dynamics of host-BVDV-1 interaction from a transcriptomic and proteomic approaches. This study shows an innovate direction trough the analysis of virus-host interaction based on the correlation between DEG and PEG analysis. The MS is well-written and organized, but some points can be improved.
Strengths: The promising results and exhaustive description of the immune genes that appear regulated in the infection condition.
[Reply] Thank you very much for your positive comments and reasonable suggestions.
We have made detailed revisions to the manuscript according to your suggestions.
Minor comments:
1. Please italicize “p” in "p < 0.05" in the manuscript.
[Reply] I have revised and marked them in green. Line 359 and 456.
2. Substitute h.p.i. by hpi.
[Reply] I have revised and marked them in green.
3. Material and methods: Could you please clarify if you used DNAse treatment in the procedure of 2.4 and 2.5?
[Reply] Yes. In the process of extracting RNA, we add DNase for purification after the first rinse.
4. Material and methods: Can you confirm if you used only one housekeeping gene and why? Next time take into account to use at least two for a more accurate analysis.
[Reply] Thank you very much for your suggestion. Literatures similar to our study only used ACTB as a housekeeping gene[1,2], so we selected a housekeeping gene. In subsequent studies we will consider using more than two housekeeping genes to obtain more accurate results.
5. Results: Please correct the quality of the figures.
[Reply] Thank you for your suggestion. Due to space limitations, some images may not be clear enough. We have uploaded higher-resolution original images in the attachment. These images can be selected during typesetting to make it easier for readers to obtain information.
6. Results: Why in the figure 1B does the mock condition appears only from 9 hpi onwards? And substitute Time(hpi) by Time (hpi).
[Reply] We found that in the kinetic model of BVDV infection of PBMCs, the infectious amount of virus was highest at 9 hpi. Therefore, we started to detect the BVDV copy number of mock-infected PBMCs from 9 hours.
7. Results: Why is q < -0.5 used as a criterion for DEGs? And FC<0.83 for DEPs?
[Reply] The amount of data differs between proteomics and transcriptomics, and so do the screening criteria. This can be found in some other articles as well[3].
8. Table 3: Fix symbols (Log2 and -)
[Reply] I have revised and marked them in green. Line 517-518
9. Line 48: Substitute Interleu-kin-17 by Interleukin-17.
[Reply] I have revised and marked them in green. Line 48
10. Line 49: Space after interaction is missing.
[Reply] I have revised it.
11. Line64-66: Rewrite the sentence because it is not clear.
[Reply] I have rewritten this sentence and marked it green. Line 64-67
12. Line 163: Space after 12 is missing.
[Reply] I have checked and corrected these errors throughout the text.
References:
- Liu, C.; Liu, Y.; Liang, L.; Cui, S.; Zhang, Y. RNA-Seq based transcriptome analysis during bovine viral diarrhoea virus (BVDV) infection. Bmc Genomics 2019, 20, 774, doi:10.1186/s12864-019-6120-4.
- Ma, Y.; Wang, L.; Jiang, X.; Yao, X.; Huang, X.; Zhou, K.; Yang, Y.; Wang, Y.; Sun, X.; Guan, X., et al. Integrative Transcriptomics and Proteomics Analysis Provide a Deep Insight Into Bovine Viral Diarrhea Virus-Host Interactions During BVDV Infection. Front Immunol 2022, 13, 862828, doi:10.3389/fimmu.2022.862828.
- Duan, F.; Wang, X.; Wang, H.; Wang, Y.; Zhang, Y.; Chen, J.; Zhu, X.; Chen, B. GDF11 ameliorates severe acute pancreatitis through modulating macrophage M1 and M2 polarization by targeting the TGFbetaR1/SMAD-2 pathway. Int Immunopharmacol 2022, 108, 108777, doi:10.1016/j.intimp.2022.108777.

Reviewer 2 Report
Line 32 is correct BVDV but not really explained, Mucosal Disease is associated with primary infection ncp BVDV infecting the fetus before about 125 days gestation resulting in a persistently infected animal that is then infected with a cytopathic BVDV that is closely related to the original ncp resulting in persistent infection.
Line 65-66 states PI animals lack antibodies to BVD is not correct, they can produce antibodies to BVDV that varies from the BVDV that caused the PI. The paper is well written, but not an easy read due to the depth of immunology covered, but this is not a demeaning the manuscript which contributes to the body of knowledge concerning BVDV. The statement in line 120 concerning establishment of a theoretical foundation for unraveling the mysteries of bvdv infection within host cells describes the paper.
The quality of English language is good.
Author Response
Response to reviewer 2
Dear reviewer 2,
Thank you very much for your a large of helpful suggestions for my manuscript. I have carefully revised the manuscript in accordance with your suggestions, the details are as follows.
Best regards,
Sincerely,
Kang Zhang
General comments:
The paper is well written, but not an easy read due to the depth of immunology covered, but this is not a demeaning the manuscript which contributes to the body of knowledge concerning BVDV. The statement in line 120 concerning establishment of a theoretical foundation for unraveling the mysteries of bvdv infection within host cells describes the paper.
[Reply] Thank you very much for your recognition of our work. In future work, we will try our best to improve our writing skills to make it easier for readers to understand our ideas.
Specific comments:
Line 32 is correct BVDV but not really explained, Mucosal Disease is associated with primary infection ncp BVDV infecting the fetus before about 125 days gestation resulting in a persistently infected animal that is then infected with a cytopathic BVDV that is closely related to the original ncp resulting in persistent infection.
[Reply] Thank you very much for your explanation of mucosal disease caused by BVDV, which gave me a clear understanding of the relationship between the two.
Line 65-66 states PI animals lack antibodies to BVD is not correct, they can produce antibodies to BVDV that varies from the BVDV that caused the PI.
[Reply] Thank you very much for your correct and professional suggestion. Searching the literatures, we did realize that the description of persistent infection (PI) in the manuscript was incorrect. I have revised these sentences and marked them in green. Line 64-67
